# Cuff-Less Blood Pressure Prediction Based on Photoplethysmography and Modified ResNet

**DOI:** 10.3390/bioengineering10040400

**Published:** 2023-03-24

**Authors:** Caijie Qin, Yong Li, Chibiao Liu, Xibo Ma

**Affiliations:** 1Institute of Information Engineering, Sanming University, Sanming 365004, China; 2CBSR&NLPR, Institute of Automation, Chinese Academy of Sciences, Beijing 100049, China; 3School of Artificial Intelligence, University of Chinese Academy of Sciences, Beijing 100049, China

**Keywords:** blood pressure, continuous prediction, ResNet34, multi-scale feature extraction, channel attention

## Abstract

Cardiovascular disease (CVD) has become a common health problem of mankind, and the prevalence and mortality of CVD are rising on a year-to-year basis. Blood pressure (BP) is an important physiological parameter of the human body and also an important physiological indicator for the prevention and treatment of CVD. Existing intermittent measurement methods do not fully indicate the real BP status of the human body and cannot get rid of the restraining feeling of a cuff. Accordingly, this study proposed a deep learning network based on the ResNet34 framework for continuous prediction of BP using only the promising PPG signal. The high-quality PPG signals were first passed through a multi-scale feature extraction module after a series of pre-processing to expand the perceptive field and enhance the perception ability on features. Subsequently, useful feature information was then extracted by stacking multiple residual modules with channel attention to increase the accuracy of the model. Lastly, in the training stage, the Huber loss function was adopted to stabilize the iterative process and obtain the optimal solution of the model. On a subset of the MIMIC dataset, the errors of both SBP and DBP predicted by the model met the AAMI standards, while the accuracy of DBP reached Grade A of the BHS standard, and the accuracy of SBP almost reached Grade A of the BHS standard. The proposed method verifies the potential and feasibility of PPG signals combined with deep neural networks in the field of continuous BP monitoring. Furthermore, the method is easy to deploy in portable devices, and it is more consistent with the future trend of wearable blood-pressure-monitoring devices (e.g., smartphones and smartwatches).

## 1. Introduction

The prevalence and mortality of cardiovascular disease (CVD) are rising on a year-to-year basis. Hypertension, as the most-prevalent chronic disease, has been confirmed as the main cause of cardiovascular and cerebrovascular diseases. A recent study suggested that the number of people with hypertension worldwide is nearly 1 billion and will increase to 1.56 billion by 2025 [1]. Besides, hypertension develops insidious early symptoms. If not treated in time, abnormal blood pressure (BP) can cause damage to the heart, brain, kidneys, retina, and other vital organs, thus resulting in severe consequences (e.g., stroke, myocardial infarction, and kidney failure).

Patients with hypertension have poor compliance with self-monitoring due to the insidious nature of hypertensive symptoms and the lack of patients’ attention. However, daily blood pressure monitoring and self-management of patients are vital tools to curb the progression of hypertension disease. Blood pressure monitoring and timely intervention are capable of significantly reducing the incidence of cardiovascular disease complications and the risk of death.

Conventional blood pressure detection methods comprise arterial catheterization, auscultation, oscillometry, volume compensation, and so forth. Arterial catheterization is an invasive method of blood pressure detection and is considered the international gold standard [2]. This method is detected by inserting a sensor probe or a catheter connected to a pressure sensor directly into an arterial vessel. Although this method is capable of obtaining continuous and accurate blood pressure, the operation is complicated, and patients are subjected to great damage, so it applies to the monitoring and treatment of severe patients in hospitals. The auscultation method employs cuff pressure to occlude the artery, and then, the pulse sounds during the cuff deflation are recorded to obtain the blood pressure [3]. In general, the principle of oscillometry is consistent with that of auscultation, except that oscillometry obtains blood pressure by examining pressure through a pressure sensor within the cuff [4]. Both of the methods are intermittent measurement methods with a simple operation, whereas the inflation and deflation operations in the measurement process will cause discomfort to the patients. Besides, they do no apply to neonates, subjects with skin lesions, or large arm circumferences. The volume compensation method makes up for the changes of blood vessel volume through servo-controlled cuff pressure, and it detects blood pressure by examining the pressure within the cuff [5]. Although this method is capable of conducting continuous measurement, the hardware system is complex and costly, and the blood vessel will be deformed under prolonged pressure. As a result, the accuracy of blood pressure measurement is affected, and the subjects are subjected to discomfort.

Blood pressure refers to a dynamic physiological parameter with a pattern of diurnal variation [6]. It is susceptible to fluctuations caused by emotional changes or external stimuli [7]. Intermittent blood pressure measurements do not fully indicated an individual’s physiological or pathological status. Noninvasive continuous blood pressure measurement is capable of eliminating the damage caused by invasive blood pressure measurement, and it gets rid of the cuff restraint caused by intermittent blood pressure measurement. Moreover, it can comprehensively assess blood pressure status, such that cardiovascular and cerebrovascular diseases can be prevented and treated.

Photoplethysmography (PPG) refers to a non-invasive optical bioassay technique that uses optoelectronic technology to record the variation of blood volume in the subcutaneous blood vessels [8]. The PPG sensor only requires a light-emitting diode to emit a beam with a certain wavelength and a photodetector to detect the attenuated beam. On that basis, it has the characteristics of a simple structure, low cost, non-invasive, and portability, which is suitable for low-power wearable devices. The blood volume in the blood vessel presents pulsating changes under the action of the heart beat. Thus, the PPG signal contains a considerable amount of cardiovascular physiological information, and it has been adopted to evaluate a wide variety of physiological parameters (e.g., blood oxygen, heart rate, respiratory rate, and blood glucose) [9,10,11,12]. Furthermore, PPG has been studied in the field of continuous blood pressure measurement [13,14]. Existing research has suggested that the PPG signal contains factors related to blood pressure, and it is a promising candidate for continuous blood pressure estimation.

## 2. Related works

### 2.1. Continuous BP Prediction Methods Based on Pulse Transit Time

Studies have demonstrated a correlation between blood pressure and pulse wave velocity (PWV), which can be calculated by the pulse transit time (PTT) [15]. The PTT can be obtained by simultaneous detection of two PPG signals from different arterial sites or by simultaneous acquisition of PPG and electrocardiogram (ECG) signals and then determining the time interval between the propagation of the two signals.

Kim et al. [16] measured two PPG signals from different sites of the finger, and the time interval between the peaks of the two signals was used to calculate the blood pressure by a regression formula. The proposed method was experimented on data collected from 21 subjects and achieved an error rate of nearly 5%. Sagirova et al. [17] calculated the time interval between the wave peak of PPG and the R-peak of ECG and obtained systolic blood pressure (SBP) and diastolic blood pressure (DBP) based on a linear regression method. In their research, the data were collected from 512 patients with a history of hypertension, and the experimental results yielded an error of 0.32 ± 3.63 mmHg for SBP and 0.61 ± 2.95 mmHg for DBP.

In the PTT-based blood pressure prediction method, two simultaneous physiological signals and wearing multiple sensors should be acquired, such that the complexity of the devices is increased. Besides, the two signals should be perfectly aligned, thus increasing the difficulty of the method.

### 2.2. Continuous BP Prediction Methods Based on Feature Extraction of PPG

In continuous BP prediction methods based on feature extraction of PPG, the morphological features in the PPG signal should be extracted in advance, and a blood pressure prediction model is built using artificial intelligence methods. Several means (e.g., introducing PTT parameters, combing with ECG signals, calibration) are adopted in many studies to increase the prediction accuracy of blood pressure.

Chen et al. [18] extracted 14 features (e.g., PTT, heart rate, and K value) from the PPG and ECG signals of the MIMIC public dataset. In their research, the support vector machine regression method was adopted, combined with mean influence value (MIV) feature selection and a genetic algorithm (GA) parameter optimization method. There were 772 sets of waveform data acquired online for the experiment, which resulted in an error of 3.27 ± 5.52 mmHg for SBP and 1.16 ± 1.97 mmHg for DBP. Thambiraj et al. [19] extracted 43 features (e.g., PTT, Womersley number, ECG-based features, and PPG-based features) from a public dataset. In their research, five common machine learning methods were compared, and the random forest regression model combined with feature selection achieved the optimal performance with errors of 9.54 and 5.48 mmHg for SBP and DBP, respectively. Furthermore, El-Hajj et al. [20] extracted 52 features from the PPG signal and its derivatives, which were subsequently reduced to 24 dimensions by feature selection. In their research, a variant of the RNN model that comprised LSTM and GRU was adopted, and the errors of 4.51 ± 7.81 mmHg for SBP and 2.6 ± 4.41 mmHg for DBP were generated on the public dataset, respectively.

This type of method is dependent on complex feature engineering, which is significantly affected by the quality of the PPG signals and the accuracy of the location of the fiducial points. Besides, feature extraction requires considerable prior knowledge, such that the model exhibits insufficient generalization performance. For instance, the PPG waveform of some subjects does not have a dicrotic wave, such that the characteristic parameters regarding the dicrotic wave cannot be generalized to all PPG signals.

### 2.3. Continuous BP Prediction Method Based on Deep Learning

Compared with the methods based on feature extraction, deep learning methods do not show the disadvantages of feature engineering. Besides, they are capable of learning more abstract and high-dimensional features from raw PPG signals and exhibit stronger modeling ability for complex nonlinear systems.

Sadrawi et al. [21] predicted the continuous arterial blood pressure (ABP) wave using single PPG signals based on a deep neural network model. In their research, the IntelliVue Patient Monitor was adopted to collect the data from 18 patients. As indicated by the results, the UNet model achieved better prediction results for SBP, whereas the Lenet-5 model obtained better prediction results for DBP. Baker et al. [22] built a CNN-LSTM model to continuously predict blood pressure based on ECG and PPG signals. The method used a CNN to extract features followed by LSTM to model the sequential data and obtained a low mean absolute error on the MIMIC public dataset. Hu et al. [23] built a neural network that incorporated multi-scale and multi-task learning to continuously estimate blood pressure. The network could acquire multi-scale features by introducing depthwise separable convolution and attention mechanisms, and it mined task-relevant features through multi-task learning. The accuracy of DBP reached Grade A and that of SBP reached grade B on the UCI dataset in accordance with the British Hypertension Society standard.

In conclusion, cuff-less continuous blood pressure monitoring is of great significance for the management of the hypertensive population and the prevention of cardiovascular diseases. The PPG signal is easily accessible, breaks through the limitation of the cuff, and contains rich physiological information regarding the vascular system, making it ideal for deployment on mobile devices such as smartphones and smartwatches. In this study, a residual network integrating multi-scale feature extraction and the channel attention mechanism is proposed for continuous blood pressure monitoring using only the raw PPG waveform as the input. The proposed method does not require complex feature engineering or redundant equipment to extract multi-site signals or ECG signals. The proposed model was validated on a large public dataset, yielding competitive blood pressure prediction results.

## 3. Materials and Methods

### 3.1. Dataset

The MIMIC dataset refers to a public multi-parameter intensive care database developed by the Massachusetts Institute of Technology [24]. The dataset used in our study is from the University of California, Irvine (UCI) Machine Learning Repository, which is a subset of the MIMIC dataset [25,26]. The UCI blood pressure dataset contains 12,000 data records, each of which includes physiological parameters such as ECG, PPG, ABP, and respiratory signals, with a sampling frequency of 125 Hz. The ABP in the UCI dataset was collected by invasive means, from which we extracted the reference values of BP. Based on the UCI dataset, only the PPG signals and their corresponding ABP records were selected for study.

### 3.2. Pre-Processing

#### 3.2.1. Signal Filtering

The PPG signal refers to a weak electrical signal that is susceptible to noise (e.g., baseline drift and power line interference). The above noise disturbances will change the shape of the signal, such that the signal should be filtered and denoised before being analyzed. The PPG signal was filtered using a fourth-order Butterworth band-pass filter in the study. Since baseline drift and power line interference were expressed as low- and high-frequency components, respectively, on the spectrum, the cut-off frequency of the band-pass filter was set to [0.5, 8] Hz.

#### 3.2.2. Segmentation

Given the need for the input of the deep learning network, the raw PPG signal should be cut into fixed-length segments through a sliding window after filtering. The fixed-length segmentation mode is simple to operate, preserves signal diversity, and does not need to locate the fiducial points. Lastly, the PPG signals were split into 3-second segments, a length that contains multiple cycles of cardiac activity without imposing a large burden on the network model. The same operation was applied in parallel to the corresponding ABP signals.

#### 3.2.3. Outlier Removal

A series of quality screening rules was adopted to exclude unreliable data, so as to obtain high-quality PPG signals and further reduce data noise. The signals with SBP greater than 180 mmHg and DBP less than 50 mmHg were first excluded from the dataset. Second, skewness can be employed to measure the asymmetry regarding the probability distribution of a random variable. Moreover, Krishnan et al. suggested that skewness is correlated with the quality of PPG signals [27]. Accordingly, the skewness index of PPG segments was evaluated, and segments with a skewness index less than zero were excluded. The calculation method of the skewness is expressed in Equation (Equation 1).
(1)$$S_{kew}=\frac{1}{N}\sum_{i=1}^{n}{\left(\frac{x_{i}-\mu_{x}}{\sigma_{x}}\right)}^{3}$$
where $\mu_{x}$ and $\sigma_{x}$ represent the mean and standard deviation of the samples in the PPG segment, respectively; *N* denotes the number of samples in the segment.

Moreover, since a high-quality PPG signal should maintain a good periodicity, the autocorrelation coefficient was adopted to evaluate the signal quality [28]. The autocorrelation coefficient was set to an empirical threshold of 0.6 in the experiments, such that the damaged PPG signal segments were excluded.

After the above pre-processing steps, 47,964 high-quality PPG signal segments were included in the final experiment. The blood pressure distribution of the final dataset is shown in Figure 1. The first-order derivative velocity plethysmography (VPG) and the second-order derivative acceleration plethysmography (APG) of the PPG signal also contain rich physiological information related to blood pressure [29]. Thus, the PPG signal and its derivatives were incorporated into the input dataset to enrich the information of the input data. Before being fed into the model, the data should be normalized to make the training process more stable and to increase the convergence speed. In this study, the Min–Max method was adopted to make the amplitude of the PPG signal fall in the [0, 1] interval. The equation is written as follows:(2)$$y^{{}^{\prime }}=\frac{y-y_{min}}{y_{max}-y_{min}}$$
where $y_{max}$ and $y_{min}$ represent the maximum and minimum values of the segment signals.

### 3.3. Proposed Method

He et al. [30] proposed a residual neural network that comprises multiple residual modules to solve the problem of network degradation caused by the increasing depth of the neural network. The residual module transfers input features directly to the output by means of a shortcut connection, such that the whole model only needs to learn the difference between the input and output, which improves the discriminative ability of the network while simplifying the network learning task. Its learning process is as follows: (3)H(x)=F(x)+x
where *x* denotes the input feature; H(x) represents the fit target; F(x) expresses the residual mapping. The original input *x* is directly correlated with the output through shortcut connection, such that information loss can be largely avoided and learning difficulty can be significantly reduced.

Figure 2 presents the model framework adopted in this study. Given the feature representation capability and the computational cost of the network, ResNet34 was adopted as the basic framework to build the continuous BP estimation model (MSA-ResNet) based on PPG signals. MSA-ResNet provides a simple and effective solution for continuous blood pressure estimation based on a tandem combination of residual structures, combined with multi-scale feature extraction, the channel attention mechanism, and the loss function optimization strategy.

#### 3.3.1. Multi-Scale Feature Extraction Module

A single-scale feature extraction module can only use fixed-size convolutional kernels, which may lead to insufficient feature extraction and cannot extract deep feature information well. The multi-scale feature extraction module actually samples the signal at different granularities, such that the network model captures different scales of receptive fields, and thus mines the features of different fineness. The structure of the multi-scale feature extraction (MSFE) module designed in this study is shown in Figure 2. The MSFE module contains four parallel convolutional branches, which comprise three regular-sized convolutional branches and a large-sized convolutional branch.

The size of the convolutional kernel in regular-sized convolutional branch was set to the commonly used 3, 5, and 7, respectively. In addition, a large-sized convolutional branch was introduced in this study. Inspired by the research of Ding et al. [31], the size of the large convolutional kernel was set to 13 in our experiments. Since a large-sized convolutional kernel will increase the computational cost of the model, a depthwise separable convolution was adopted rather than the regular convolution in the large-sized convolutional branch. Depthwise separable convolution can significantly reduce the number of parameters in the network by performing convolutional operations on each channel of input layer independently. Subsequently, pointwise convolution was used to adjust the channels to compensate for the loss of information exchange between channels due to the depthwise separable convolution.

Moreover, the padding operation was adopted to ensure that the output feature length of each parallel branch was consistent. The results of each branch were merged by the concatenation operation, followed by the convolutional operation to reshape the dimension and then input into the subsequent residual structures.

#### 3.3.2. Channel Attention Module

The squeeze-and-excitation (SE) module refers to an attention module based on the channel dimension [32]. It extracts useful feature information from each channel of the feature map while suppressing useless global information. The SE module consists of a squeeze operation and an excitation operation. First, the squeeze operation calculates the average of each channel in the feature map by global average pooling (GAP) to reduce the number of weight parameters applied in the subsequent fully connected layers. The calculation procedure of the squeeze operation is expressed in Equation (Equation 4).
(4)$$Z_{c}=F_{sq}\left(x_{c}\right)=\frac{1}{W}\sum_{i=1}^{W}x_{c}\left(i\right)$$
where $x_{c}$ denotes the cth channel of feature map *x*; *W* represents the width of the feature map.

Second, the excitation operation adopts two fully connected (FC) layers to first reduce the dimension and then increase the dimension, and the ReLU function serves as the activation function between the two FC layers. The output of the last FC layer is activated by the Sigmoid function to determine the attention weight of the channel. The calculation procedure of the excitation operation is expressed in Equation (Equation 5).
(5)$$s=sigmoid(FC_{2}\left(Relu\left(FC_{1}\left(z\right)\right)\right))$$
where *z* denotes the global receptive field of *c* feature channels obtained by the squeeze operation and *s* denotes the weight coefficient of *c* feature channels.

In the last step of feature weight calibration, the channel weights calculated by the SE module are multiplied with the respective corresponding channels in the feature map to obtain the effective feature layer with attention information. The calculation procedure is expressed in Equation (Equation 6).
(6)$${\tilde{x}}_{c}=F_{scale}\left(x_{c},s_{c}\right)=x_{c}\times s_{c}$$

#### 3.3.3. Loss Function Optimization

The mean absolute error (MAE) and mean-squared error (MSE) are commonly used evaluation indexes in regression tasks, and the above two types of losses are commonly used for gradient optimization during the training process of the models. Assuming that ${\tilde{y}}_{i}$ and $y_{i}$ are the predicted and reference values, respectively, the MSE and MAE are calculated as shown in Equation (Equation 7).
(7)$$MSE=\frac{1}{m}\sum_{i=1}^{m}{\left(y_{i}-{\tilde{y}}_{i}\right)}^{2}\;MAE=\frac{1}{m}\sum_{i=1}^{m}|y_{i}-{\tilde{y}}_{i}|$$

The MSE calculates the mean of the squared error between the predicted and reference values, so using the MSE loss function is convenient for the gradient descent algorithm and facilitates the convergence of the function. However, if there are outliers in the dataset, the MSE loss function will be significantly affected by the outliers. The MAE loss function is more robust to outlier points, but more complicated to compute the gradient. The Huber loss is calculated as shown in Equation (Equation 8), which is a loss function with parameters used to solve the regression problem. The Huber loss combines the advantages of the MAE loss and MSE loss, reduces the sensitivity to outlier points, avoids model overfitting to a certain extent, and achieves a better derivability of the functions.
(8)$$L_{\delta }\left(y,\tilde{y}\right)=\left\{\begin{array}{c} 0.5{\left(y-\tilde{y}\right)}^{2},if|y-\tilde{y}|\leq \delta \\ \delta |y-\tilde{y}|-0.5\delta^{2},if|y-\tilde{y}|>\delta \end{array}\right.$$
where δ is a super parameter, and its default setting is 1.

## 4. Results

The experiments were performed on an NVIDIA RTX 3090 GPU, using a software platform based on cuda 11.1, pytorch 1.10, and python 3.8. The training effect of the convolution neural network is closely related to the settings of the model parameter, so the parameters of each model were unified at the training stage. The dataset was divided into a training set, a validation set, and a test set at the ratio of 7:1:2. The batch size was set to 1024 for each iteration. The learning rate was dynamically adjusted using the SGD optimization and cosine annealing decay, with the initial learning rate set to 0.001. Furthermore, 120 epochs were trained for all models to ensure the convergence of the model.

### 4.1. Ablation Experiment

In this study, the following improvements were proposed based on ResNet34: using a multi-scale feature extraction module rather than a regular feature extraction module in the baseline network, introducing the channel attention mechanism in the residual module, and adopting the optimized Huber loss function at the training stage. To verify the effectiveness of the above improvements, ablation and comparison experiments were conducted for each of the above improvements, respectively. Table 1 lists the experiment results, where “✓” means the improvement strategy was adopted and ”×” means the improvement strategy was not adopted.

As depicted in Table 1, the baseline network ResNet34 achieved an MAE of 7.87 mmHg for SBP and 4.51 mmHg for DBP without introducing any improvement module. On that basis, the initial feature extraction module of the baseline network was replaced with the MSFE module, and the estimation errors for SBP and DBP were decreased by 0.98 mmHg and 0.62 mmHg, respectively. Notably, the performance of the model was significantly enhanced by introducing the SE attention module under the baseline network, in which the estimation error for SBP and DBP were reduced by 1.49 mmHg and 0.89 mmHg, respectively.

Furthermore, the use of the Huber loss during model training resulted in a decrease of 0.66 and 0.35 mmHg in the estimates of SBP and DBP, respectively. Incorporating all the improved modules can further lead to the enhancement of performance, and the MAEs of the final proposed model for SBP and DBP were 5.98 mmHg and 3.24 mmHg, respectively, as indicated by the experimental results.

### 4.2. Comparison with AAMI Protocol and BHS Protocol

The Association for the Advancement of Medical Instrumentation (AAMI) has established internationally accepted error standards for blood pressure [33]. The predicted SBP and DBP should have a mean error of less than ±5 mmHg and a standard deviation of less than 8 mmHg. The mean error (ME) and standard deviation (SD) are calculated as shown in Equation (Equation 9). However, since the ME and SD are susceptible to outliers, the British Hypertension Society (BHS) has devised an alternative protocol for evaluating BP prediction results using cumulative percentages of errors within the range of 5, 10, and 15 mmHg [34].
(9)$$ME=\frac{1}{m}\sum_{i=1}^{m}\left(y_{i}-{\tilde{y}}_{i}\right)\;SD=\sqrt{\frac{1}{m-1}\sum_{i=1}^{m}{\left(y_{i}-{\tilde{y}}_{i}\right)}^{2}}$$
where $y_{i}$ and ${\tilde{y}}_{i}$ denote the reference and predicted blood pressure by the model, respectively, and *m* is the number of samples.

Table 2 and Table 3 present the comparisons of the experimental results with AAMI and BHS standards. The ME and SD for both SBP and DBP met the AAMI standard. Moreover, the accuracy of the predicted DBP reached Grade A of the BHS standard, while the accuracy of the predicted SBP was very close to Grade A.

### 4.3. Statistical Analysis

The Bland–Altman plot is a common method to evaluate the consistency of two continuous variable measures. The two measurement methods used for comparison usually comprise a new method that needs to be investigated and an accepted standard method. The two methods are considered to be in good agreement if 95% of the differences between two measurement results lie within the range of (mean-1.96*SD) and (mean+1.96*SD).

Table 2 shows that the ME of the predicted SBP was 0.92 with a standard deviation of 7.79, while the ME of the predicted DBP reached 0.68 with a standard deviation of 4.94. As depicted in Figure 3, most of the samples lie between the range of the limits of agreement (LOAs) with the 95% confidence interval, suggesting that the predicted blood pressure values were very consistent with the reference values. Furthermore, Figure 4 presents the correlation between the predicted and reference values of blood pressure. The correlation coefficients were 0.773 for SBP and 0.851 for DBP, suggesting that the predicted blood pressure and the reference values showed the same trend and a significant correlation.

## 5. Discussion

Currently, continuous cuff-less blood pressure monitoring is urgently required. Given the potential of the PPG signal in blood pressure estimation, the PPG signal and deep learning network were employed in this study to estimate blood pressure continuously.

In this study, a multi-scale feature extraction module was introduced based on the basic framework of ResNet34. The initial module of the novel framework fully explored the features of the input signal by activating the multi-scale receptive field. The multi-scale feature extraction module contained regular-sized convolutional branches. Moreover, it added a large-sized convolutional branch. In the large-sized convolutional branch, a combination of depthwise separable convolution with a kernel size of 13 and pointwise convolution was used to reduce the number of parameters while increasing the effective receptive field. As depicted in Table 1, the multi-feature extraction module increased the accuracy of SBP and DBP, suggesting that the module can enhance the capability of feature extraction and representation.

Each channel of the feature map is equally important by default in the conventional convolution and pooling processes. In contrast, the importance of different channels is not the same in actual problems. The SE attention module facilitates the feature extraction of the previous layer by reassigning channel features. Moreover, short connections in the residual module are capable of significantly alleviating the problem of gradient disappearance and facilitating feature propagation. The result indicated that the embedded attention mechanism module contributed well to the prediction of SBP and DBP, suggesting that the SE attention mechanism can obtain a more discriminative feature representation, thus increasing the prediction accuracy of the model.

The derivative of the MAE loss function is constant, thus hindering gradient calculation. If the learning rate is constant, the loss function will fluctuate around the stable value, and the model may obtain a local optimal solution. The MSE loss function is susceptible to outliers. Moreover, if the outliers are damaged abnormal samples, this will affect the convergence trend of the model. The Huber loss function improves the shortcomings of the above two loss functions and is more robust to outliers, and it is easier to obtain the global optimal solution. Probably because the original PPG segments were screened during the dataset pre-processing, the quality of the dataset was improved. Accordingly, as revealed by the experimental results, we found that the performance of the model was slightly enhanced by using the Huber loss function for training.

Furthermore, as indicated by the experimental results, DBP exerted a better prediction effect than SBP. The reason for this result is that the fluctuation range of SBP was significantly larger than that of DBP, thus leading to a higher prediction error.

### Comparison Results with Related Work

Although a large number of studies have emerged in the field of continuous BP prediction, it is difficult to directly compare our experimental results with related work due to differences in the dataset size, input signals used, and evaluation metrics. Many studies adopted PPG signals combined with ECG signals, which can introduce more physiological information and increase the accuracy of the model. However, such methods also have several limitations. For instance, the above methods require two types of sensors, so they are difficult to deploy on mobile devices such as a smartphone because it is difficult to make further modifications to the hardware of a smartphone that has already been produced. Moreover, there is also difficulty in the alignment of the two signals. There are also some studies that have used features extracted from PPG signals as the inputs, which rely on strong prior knowledge. Thus, for a better comparison, we compared the proposed method with the latest studies based on the MIMIC public dataset, using only the raw PPG signals as the input. From Table 4, we find that the performance of the proposed method was superior to most of the other methods. Although the study [35] obtained a more reasonable accuracy with a smaller sample size, our work produced a slightly higher error on a larger dataset. This may be due to the fact that a larger dataset contains more diverse PPG signals, and the model has more difficulty predicting, while the performance becomes worse. In general, this study obtained reasonable BP prediction results on a large dataset only using easily accessible PPG signals. This fully demonstrates the potential and feasibility of PPG signals combined with a deep learning network in the field of continuous BP prediction.

In this study, the PPG records in the MIMIC dataset were all collected from ICU patients with severe symptoms, whose blood pressure may be affected by disease, medications, and other factors. On that basis, the above PPG signals are not representative of the signal characteristics of most people. Besides, by a comprehensive analysis of Figure 1, Figure 3 and Figure 4, we found that some of the larger errors are concentrated in regions with smaller sample sizes (e.g., DBP > 80, SBP > 170, and SBP < 100). The above regions have smaller sample sizes in the overall dataset, whereas the model is more likely to be biased toward regions with larger sample sizes during training. Therefore, a considerable number of PPG signals from different populations will be collected to enhance the diversity of the data samples in the future. Furthermore, our research team is working on the experiment of deploying the algorithm on a smartphone, which is also a very promising research task for the realization of home-based daily blood pressure monitoring.

Besides, the computational complexity is the primary issue that must be considered in practical applications. Resorting to a cloud server can be a proposed solution. Developers can deploy the proven models on cloud servers. The signal is acquired and initially preprocessed on the mobile side, then the data are transmitted to the cloud server for prediction, and the results are returned to the mobile side. With the communication between cloud servers and wearable devices, the problem of the insufficient computing power of wearable devices can be solved. Figure 5 shows the frame diagram of the communication between the cloud server, client, and web side. Using cloud servers to deploy projects is the first option for many developers, and there are some mature frameworks (e.g., tensorflow, pytorch, keras, caffe, theano, mxnet) that can be used. In addition, using cloud servers for computation is highly performant and can handle relatively large models. However, this approach relies on the Internet speed, and the real-time requirements of the project need to be evaluated. Since blood pressure does not change rapidly in the resting state, deploying the proposed algorithm on a cloud server can satisfy the real-time requirements.

## 6. Conclusions

A deep learning network for continuous blood pressure prediction using only PPG signals was proposed in this study. Under the ResNet34 framework, the proposed network replaced the initial feature extraction module with a multi-scale feature extraction module and introduced the channel attention mechanism in the subsequent residual module. Moreover, a comprehensive and robust Huber loss function was adopted in the training stage. The proposed method yielded competitive prediction accuracy on the UCI dataset. The prediction errors of SBP and DBP conformed to the AAMI standard, and the accuracy of DBP reached Grade A of the BHS standard, while the accuracy of SBP almost reached Grade A. The proposed method only used PPG signals which is feasible and promising. It did not require the simultaneous acquisition of two signals or complex feature engineering. Thus, a single-channel PPG signal combined with an end-to-end deep learning network for continuous blood pressure prediction serves as a simple and effective solution for blood pressure prediction on portable wearable devices in the future.

## Figures and Tables

**Figure 1 bioengineering-10-00400-f001:**
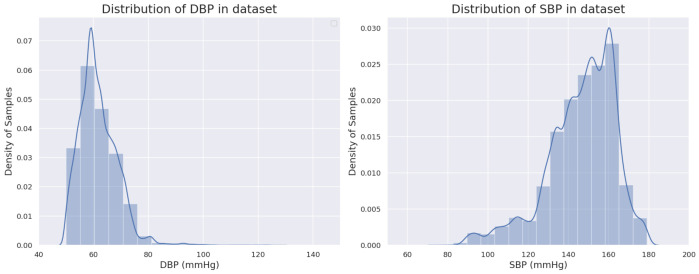
Distribution of BP in the dataset.

**Figure 2 bioengineering-10-00400-f002:**
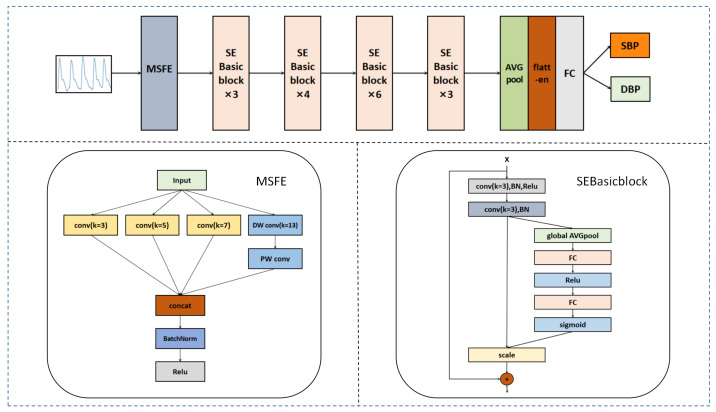
The framework of MSA-ResNet.

**Figure 3 bioengineering-10-00400-f003:**
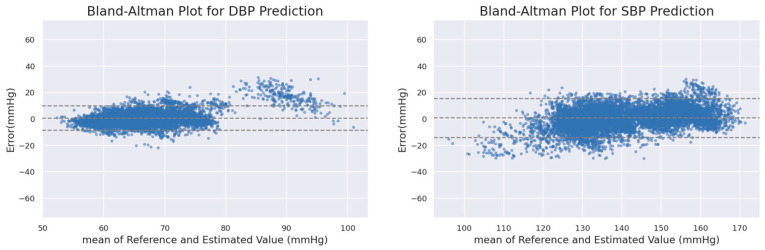
Bland–Altman plots for BP prediction.

**Figure 4 bioengineering-10-00400-f004:**
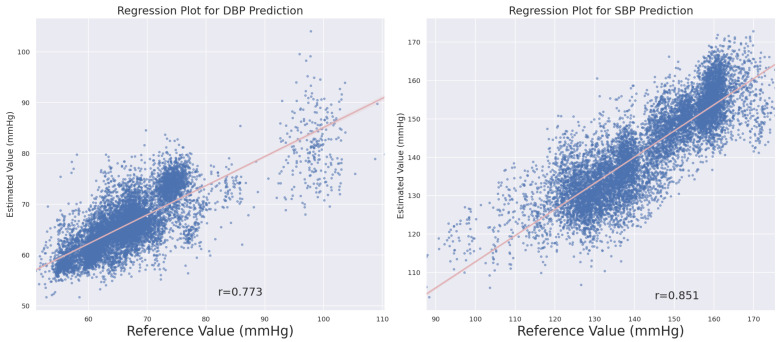
Regression plots for BP prediction.

**Figure 5 bioengineering-10-00400-f005:**
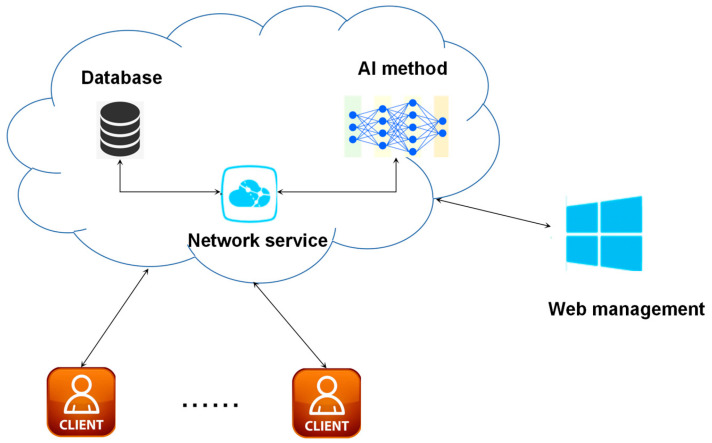
Framework of the cloud server.

**Table 1 bioengineering-10-00400-t001:** Results of the ablation experiment (mmHg).

Method	Baseline	MSFE	SE -Attention	Huber Loss	SBP	DBP
					MAE	MAE
ResNet34	✓				7.87	4.51
ResNet34 +MSFE		✓			6.89	3.89
ResNet34 +SE_Attention			✓		6.38	3.62
ResNet34 +Huber Loss				✓	7.21	4.16
Proposed Method	✓	✓	✓	✓	5.98	3.24

**Table 2 bioengineering-10-00400-t002:** Comparison with the AAMI protocol.

		ME (mmHg)	SD (mmHg)
AAMI	BP	≤5	≤8
Proposed Method	DBP	0.68	4.94
	SBP	0.92	7.79

**Table 3 bioengineering-10-00400-t003:** Comparison with the BHS protocol.

Cumulative Error (%)				
		≤**5 mmHg**	≤**10 mmHg**	≤**15 mmHg**
BHS	Grade A	60%	85%	95%
	Grade B	50%	75%	90%
	Grade C	40%	65%	85%
Proposed Method	DBP	80.8%	94.7%	98.1%
	SBP	52.1%	82.3%	93.6%

**Table 4 bioengineering-10-00400-t004:** Comparison with related work (mmHg).

Authors	Dataset	Model	SBP	DBP
Slapnicar et al. [36]	MIMIC III	ResNet	MAE = 9.43	MAE = 6.88
Li et al. [35]	MIMIC II	GRNN	MAE = 3.96	MAE = 2.39
Paviglianiti et al. [37]	MIMIC	ResNet+LSTM	MAE = 7.122	MAE = 3.534
Schrumpf et al. [38]	MIMIC III	AlexNet	MAE = 8.8	MAE = 4.9
Schrumpf et al. [38]	MIMIC III	ResNet	MAE = 7.7	MAE = 4.4
Proposed Method	MIMIC II	MSA-ResNet	MAE = 5.98	MAE = 3.24

## Data Availability

The original data are available from the corresponding author upon appropriate request.

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
