# Peer review of "Cuff-Less Blood Pressure Prediction Based on Photoplethysmography and Modified ResNet"

_bioengineering, 2023, doi:10.3390/bioengineering10040400_

Round 1
Reviewer 1 Report
In the manuscript, the author proposed a deep learning network based on the ResNet34 framework using PPG signals. And the result of the proposed method shows high prediction accuracy. The manuscript is well organized with detailed discussion. However, minor revision is needed before it can be published. Following are the details of suggestions:
1. Can the author explain more about why the error of estimation is much larger in the 80-90 mmHg range of SBP compared with the rest of DBP in Figure 3? Can the author improve that?
2. Can the author show the MAE of SBP and DBP using MIMIC II dataset? This will help to show the superiority of the proposed method over the GRNN method.
3. Can the author switch the position of the two figures in Figure 1 since all the rest of the figures present the data of DBP first except Figure 1? This will confuse readers and cost extra time to go back and forth.
Author Response
Response to Reviewer 1 Comments
In the manuscript, the author proposed a deep learning network based on the ResNet34 framework using PPG signals. And the result of the proposed method shows high prediction accuracy. The manuscript is well organized with detailed discussion. However, minor revision is needed before it can be published.
Thank you very much for your review. We are very grateful for your concerns and suggestions. The detailed responses for yours are listed point to point as follows, and the revised version has all its changes clearly marked in the manuscript.
Point 1: Can the author explain more about why the error of estimation is much larger in the 80-90 mmHg range of SBP compared with the rest of DBP in Figure 3? Can the author improve that?
Response 1: Thanks for the comment and valuable suggestion. We did find this problem in our experiments that the error of estimation is much larger in the 80-90 mmHg range of DBP. First we carefully investigated the reasons and excluded the deviations caused by the proposed algorithm. We found a very small number of DBP samples with 80-90 mmHg in the dataset, which is also reflected in the distribution of sample data in Figure 1. As the algorithm tends to pay more attention to categories with a large number of samples, the relationship between data and label values cannot be well learned in the regions with small sample size. We have tried modeling with other deep learning models, and unfortunately this part of the sample still has a large error. This issue reminds us to take into full consideration the balance and diversity of the data when constructing the data set in the next step. It is foreseeable that this problem will be better solved when the data set sample is more balanced.
Point 2: Can the author show the MAE of SBP and DBP using MIMIC II dataset? This will help to show the superiority of the proposed method over the GRNN method.
Response 2: Thank you very much for your suggestion. In this work, the Physionet’s Multi-parameter Intelligent Monitoring in Intensive Care (MIMIC) II (version 3, accessed on Sept. 2015) online waveform database [25,26] is used as a source for the PPG signals. We are very sorry for the confusion caused by our lack of detailed labeling. We have revised it in the manuscript. Although we used the same data source as the work in Table 4, the size of the data set may vary due to different data cleaning standards. It is true that the work of Li et al. obtained more competitive results in terms of evaluation metrics, but the final data set consists of 9549 signals. In contrast, our work is carried out on a much larger data set which consists of 47,964 signals, and the results obtained are more confirmatory.
[25]Kachuee, M.; Kiani, M.M.; Mohammadzade, H.; Shabany, M. Cuffless Blood Pressure Estimation Algorithms for Continuous Health-Care Monitoring. IEEE Trans Biomed Eng. 2017, 34, 859–869.
[26]Kachuee, M.; Kiani, M.M.; Mohammadzade, H.; Shabany, M. Cuff-less high-accuracy calibration-free blood pressure estimation using pulse transit time. In Proceedings of IEEE International Symposium on Circuits and Systems (ISCAS), Lisbon, PORTUGAL, 24-27 May 2015.
Point 3: Can the author switch the position of the two figures in Figure 1 since all the rest of the figures present the data of DBP first except Figure 1? This will confuse readers and cost extra time to go back and forth.
Response 3: Thanks for your helpful suggestion. It has been modified in the revised manuscript and is highlighted in red.

Reviewer 2 Report
The manuscript presents a study on the accuracy of deep learning in predicting blood pressure starting from a single PPG signal.
The obtained results show a final accuracy generally better than other state-of-the-art approaches available in the literature.
The treated topic is very timely and of high interest.
The two main observations which came to my mind while reading the paper where the necessity to test the algorithm on different data sets including "normal" subjects (as opposed to ICU recordings) and the possibility of implementing the proposed network on a mobile platform.
I saw that the authors have identified these same issues and cited them as future developments.
I agree that testing the deep learning algorithm on different dataset can be the subject of future work, but I think that some more discussions are warranted on the topic of the algorithm implementation.
I do not feel comfortable about the possibility to implement the technique on a low-resource platform since the test have been carried out employing an NVIDIA GPU. By the way, the authors mention that they are working towards implementation of the algorithm on a smartphone, but even this might be too optimistic. Most of the wearable devices which acquire PPG signals do not transmit raw PPG data, thus requiring the processing to be performed on the wearable device itself, where the computational resources might be extremely limited even compared with the processing power of a modern smartphone.
The topic of computational complexity is crucial to understand if the proposed technique is a viable solution for performing wearable real-time blood pressure monitoring, or if it is restricted to off-line processing of PPG recordings.
Please also note that in the lower left panel of Fig. 2 "MSFF" should be replaced with "MSFE".
Author Response
Response to Reviewer 2 Comments
The manuscript presents a study on the accuracy of deep learning in predicting blood pressure starting from a single PPG signal.
The obtained results show a final accuracy generally better than other state-of-the-art approaches available in the literature.
The treated topic is very timely and of high interest.
Thank you very much for your review. We are very grateful for your concerns and suggestions. The detailed responses for yours are listed point to point as follows, and the revised version has all its changes clearly marked in the manuscript.
Point 1: The two main observations which came to my mind while reading the paper where the necessity to test the algorithm on different data sets including "normal" subjects (as opposed to ICU recordings) and the possibility of implementing the proposed network on a mobile platform.
I saw that the authors have identified these same issues and cited them as future developments.
Response 1: Thank you very much for your suggestion. You mentioned the validation of the algorithm on different datasets and the application on mobile platforms, which is really what we are more concerned about in the future research direction. As we described in the paper, blood pressure is a dynamic physiological parameter that is susceptible to fluctuations due to external factors such as emotions and stress, and therefore there is a need for dynamic blood pressure monitoring in the general population of subjects. In addition, the deployment of algorithms on mobile platforms will be a big trend in the future, which will make it more convenient for users to monitor blood pressure . These above elements will be the future direction of our work.
Point 2: I agree that testing the deep learning algorithm on different dataset can be the subject of future work, but I think that some more discussions are warranted on the topic of the algorithm implementation.
I do not feel comfortable about the possibility to implement the technique on a low-resource platform since the test have been carried out employing an NVIDIA GPU. By the way, the authors mention that they are working towards implementation of the algorithm on a smartphone, but even this might be too optimistic. Most of the wearable devices which acquire PPG signals do not transmit raw PPG data, thus requiring the processing to be performed on the wearable device itself, where the computational resources might be extremely limited even compared with the processing power of a modern smartphone.
Response 2: Your suggestions are very helpful to us. Although deep learning algorithms have made great progress in the field, applying algorithms on low-resource platforms is really the first issue we need to consider and the problem we have to solve in practical applications. For smart watches, PPG signals are relatively easy to obtain, and we can realize the deployment of PPG signal acquisition equipment to smart watches. For smartphones, we really don't have the ability to make changes to its hardware. We can extract the PPG signal by means of finger-end video, and this part of the work is being carried out by our team. Since deep learning algorithms do require high computing power, this is a serious problem for both wearable devices and smartphones. To deal with this problem, our solution is to use the cloud server, which can facilitate the deployment of the model, and realize the blood pressure estimation by using the network communication between the device and the cloud server. It's a great honor to communicate with you and your suggestions are very helpful to us.
Point 3: The topic of computational complexity is crucial to understand if the proposed technique is a viable solution for performing wearable real-time blood pressure monitoring, or if it is restricted to off-line processing of PPG recordings.
Response 3: Thank you for your meaningful suggestions. The achievements of deep learning on big data sets are indeed evident to all, including in the biomedical field as well. With the increasing number of people suffering from hypertension and the various complications and even serious life-threatening symptoms caused by hypertension, we hope to solve the urgent problem of blood pressure monitoring in a more convenient way. Therefore, the deployment of blood pressure monitoring to wearable devices will be a very attractive work in the future. The computational complexity you mentioned is the primary issue that must be considered in practical applications. We can package the model and deploy it on the cloud server. With the communication between the smartphone and the cloud server, the problem of insufficient computing power of wearable devices can be solved. In addition, the blood pressure parameters do not change rapidly in the resting state, so deploying the algorithm on a cloud server can meet the real-time requirements. We will also continue to streamline and optimize our models, making better trade-offs between model complexity and real-time performance.
Point 4: Please also note that in the lower left panel of Fig. 2 "MSFF" should be replaced with "MSFE".
Response 4: Thank you for pointing this out. It has been modified in the revised manuscript and is highlighted in red.

Round 2
Reviewer 2 Report
I appreciate from the authors' response that they are aware of the difficulty in implementing deep learning schemes on wearable devices and mobile phones. I also believe that the proposed solution of resorting to the cloud is a viable one.
However, I do not see any hint to the solution of using a cloud server in the revised text, where the solution of developing the algorithm on a smartphone is, instead, still mentioned.
Author Response
Point 1: However, I do not see any hint to the solution of using a cloud server in the revised text, where the solution of developing the algorithm on a smartphone is, instead, still mentioned.
Thank you for pointing this out. We have added this aspect to the manuscript and highlighted it in red. Thank you for your meaningful suggestions again.
“Besides, the computational complexity is the primary issue that must be considered in practical applications. Resorting to the cloud server can be a proposed solution. Developers can deploy the proven models on cloud servers. With the communication between cloud servers and wearable devices, the problem of insufficient computing power of wearable devices can be solved.”